# Convolutional Neural Network-Based Rapid Post-Earthquake Structural Damage Detection: Case Study

**DOI:** 10.3390/s22176426

**Published:** 2022-08-25

**Authors:** Edisson Alberto Moscoso Alcantara, Taiki Saito

**Affiliations:** Department of Architecture and Civil Engineering, Toyohashi University of Technology, Toyohashi 441-8580, Aichi, Japan

**Keywords:** convolutional neural network, power wavelet spectrum, damage detection, structural health monitoring

## Abstract

It is necessary to detect the structural damage condition of essential buildings immediately after an earthquake to identify safe structures, evacuate, or resume crucial activities. For this reason, a CNN methodology proposed to detect the structural damage condition of a building is here improved and validated for two currently instrumented essential buildings (Tahara City Hall and Toyohashi Fire Station). Three-dimensional frames instead of lumped mass models are used for the buildings. Besides this, a methodology to select records is introduced to reduce the variability of the structural responses. The maximum inter-storey drift and absolute acceleration of each storey are used as damage indicators. The accuracy is evaluated by the usability of the building, total damage condition, storey damage condition, and total comparison of the damage indicators. Finally, the maximum accuracy and R^2^ of the responses are obtained as follows: for the Tahara City Hall building, 90.0% and 0.825, respectively; for the Toyohashi Fire Station building, 100% and 0.909, respectively.

## 1. Introduction

The damage condition of essential buildings immediately after an earthquake is one of the most critical indicators for future decision-making by the government, owners, and stakeholders. Acceleration or displacement recordings from instrumented buildings during earthquakes offer valuable information to identify and monitor their damage extent. Thus, structural health monitoring is an expanding field that allows for establishing procedures to screen the structural status of buildings. For example, in order to tackle federal buildings, Mehmet Çelebi from the United States Geological Survey (USGS) reported guidelines for the seismic instrumentation of structures as part of a USA project in 2002 [1]. On the other hand, the structural information obtained from the building after an earthquake can be used for different purposes, which depend on the detail level. For instance, the usability of the buildings related to protecting their occupants is needed immediately, and a deep behavioral understanding of the structure is obtained after months [2]. However, the current conventional post-earthquake actions report the usability of buildings in days or even weeks after the event [3].

Structural responses such as the maximum inter-storey drift and acceleration can be used in order to determine structural integrity. Hazus is a geographic information system-based natural hazard analysis tool, used by the Federal Emergency Management Agency of the USA, and the Hazus earthquake model evaluates the damage probability of buildings and infrastructures considering inter-storey drift and acceleration limits to establish the structural and nonstructural damage states [4].

The accuracy of the structural responses depends mainly on the adopted structural model and the type of structural analysis. For example, a lumped mass model (LMM) is less accurate than a three-dimensional frame model (3D-FM) of high-rise or irregular structural configuration buildings. Nevertheless, the computation time with an LMM is much shorter. To improve the convergence and computational speed of building structural analyses, Koh et al. proposed an improved condensation-based method for 3D-FM. Besides this, Yoon et al. proposed a methodology to determine the LMM parameters via nonlinear analyses of 3D-FM, and applied it to an irregular structural configuration and high-rise building [5].

Recently, Machine Learning (ML) has been studied to assess damage without response analysis of structures. ML is the part of artificial intelligence that uses statistical methods to obtain experience (learning process) from main features to predict future actions or responses. An accurate prediction is obtained if the number of features increases [6]. Deep Learning (DL) is a subset of ML inspired by the work of neurons in a brain. DL makes a multi-layer neural network computation to perform this task [7]. The Convolutional Neural Network (CNN) is a part of DL that works by analyzing visual imagery. Since CNN can manage a large amount of data through the pixels of the images, its accuracy is commonly higher than that of other ML methods. For example, Hasan et al. developed a comparative analysis to classify vegetation species using three ML methods: the support vector machine (SVM), artificial neural network (ANN), and CNN, and the accuracy obtained was 91%, 94%, and 99%, respectively [8]. Keeling et al. compared CNN with ML methods for text classification, including Logistic Regression, SVM, and Random Forest. The precision rate of 75% showed that CNN performs slightly better than others on average [9]. Jiang et al. compared the performance in image classification of capsule network (CapsNet), CNN, and fully convolutional network (FCN) methods, concluding in general that the CNN and FCN models obtained a better performance than CapsNet [10].

Recently, studies have been conducted in the field of structural engineering to develop CNN models that use the time domain responses as input data. For instance, Zhang et al. developed a physics-guided convolutional neural network (PhyCNN) for data-driven structural seismic response modeling. The proposed PhyCNN considers the ground motion as input and the structural responses as output data to learn the feature mapping between them [11]. Moreover, Teng S. et al. used the acceleration response in a one-dimensional (1D) CNN followed by a decision-level fusion strategy to improve the accuracy of structural damage detection [12]. Other studies considered more than one structural parameter to increase the features of the input data. Park H. et al. established a CNN-based strain prediction technique that enables structural safety evaluations in cases of the absence or defect of strain sensors. The CNN model used dynamic acceleration and displacement responses as input data to predict the strains of structural members [13]. Xu Y. et al. used 48 intensity measures to represent ground motion characteristics as input data to indicate the damage state of structures [14].

On the other hand, studies have been conducted using frequency domain responses to improve the accuracy of CNN prediction. For example, Oh B.K. et al. built a CNN model using the displacement response, displacement frequency, and the wind speed frequency of tall buildings as input maps to estimate the safety of instrumented columns from their maximum and minimum strains [15]. Additionally, Liu T. et al. used the transmissibility function (ratio of the cross-spectral density and the auto-spectral density of the response) as input data for a 1D CNN to identify structural damage [16].

Time–Frequency Distribution (TFD) graphs represent time-domain and frequency-domain features of a signal from only one image. Hence, TFD could be used as an input map of the two-dimensional (2D) CNN algorithm. For instance, Ghahremani B. et al. used the fast S-transform from the structural acceleration response as an input map of a CNN model [17]. Similarly, Wang X. et al. applied the Hilbert–Huang time–frequency spectrum in a 2D CNN model to identify damage conditions of a benchmark structure [18]. Furthermore, Lu X. et al. developed a CNN-based rapid post-event seismic damage evaluation methodology using the continuous wavelet transform (CWT) to extract the time and frequency features of the ground motion acceleration [19]. In addition, Moscoso Alcantara et al. [20] developed a framework in which a 2D CNN model predicts the maximum ductility ratio, the inter-storey drift, and the maximum absolute acceleration of each storey of the LMM using the acceleration record of a single sensor located on the top floor of the building, where the wavelet spectra were obtained from the absolute acceleration of this sensor and used as images for the input maps.

In this study, the authors update and improve the damage identification method proposed by Moscoso Alcantara et al. [20] as follows:The structural models are 3D-FM. This allows all buildings to have different lateral force-resisting systems, structural configurations, material types, and elastic and inelastic behavior of their members;The validation of the CNN model is applied to two instrumented buildings in Japan;The structural responses used as damage identifiers are the maximum inter-storey drift (SD) and the maximum absolute acceleration (AA) of each storey of the target buildings;A methodology to select records for each damage identifier is introduced using the Incremental Dynamic Analysis (IDA) responses of each target building, where the ground motions are scaled in order to cover the elastic and inelastic behavior of the target building;The input map data for the training CNN model use the Wavelet Power Spectrum (WPS) computed from the absolute acceleration response measured by the sensor located on the top floor of each target building.

The accuracy of the results is evaluated by comparing the damage condition of the building with the reference values.

Although training and validating the CNN model is computationally intensive, once the CNN model is developed, the CNN algorithm trained for the target building can automatically predict the elastic and inelastic structural responses, and detect the damage condition immediately after the earthquake.

This paper contains sections as follows: In Section 2, the methodology is presented, including an overview of the proposed research procedure. Section 3 shows the general information about the target buildings. Section 4 establishes the nonlinear structural models used for the target buildings. Damage levels based on the maximum inter-storey drift and absolute acceleration as damage identifiers are defined in Section 5. Furthermore, Section 6 presents the methodology used to select records from the database to reduce the variability of structural responses. Section 7 specifies the wavelet power spectrum used as input data for the CNN model, and its procedure and characteristics are described in Section 8. Additionally, Section 9 defines the training and validation process used to obtain a trained CNN model. This is applied to the target buildings, and their prediction results are shown in Section 10. Finally, in Section 11, a summary of conclusions and a discussion of the research results are presented.

## 2. Methodology

As shown in Figure 1, there are two processes for obtaining the ML (CNN) model in this methodology. They are called the training process (TP) and the validation process (VP). Hence, the selected records (see Section 6) are divided into two parts for each process. In total, 90% of the ground motions are used for the TP, and 10% for the VP.

In the TP, there are two subprocesses, the data preparation process and the training of the model process. The input data (WPS) and the output reference (SD or AA on each floor) are obtained from the IDA in the data preparation process. The trained CNN model is obtained in the training of the model process and used in the VP.

In the VP, the data preparation process is used in order to predict new input data and output references. Subsequently, the trained CNN model (from the TP) is validated when the highest accuracy is found by comparing it to structural damage identification.

## 3. Target Buildings

Two target buildings are considered to validate a CNN model. They are instrumented buildings in Japan: Tahara City Hall (steel structural system) and Toyohashi Fire Station building (steel-reinforced concrete structural system).

### 3.1. Tahara City Hall Building

The Tahara City Hall is a local government office building located in Toyohashi city of Aichi prefecture in Japan (see Figure 2). This building is an instrumented building, and the location of the sensor is shown in Figure 3. The main structural characteristics are as follows:The structural system of the building is a moment-resisting frame in steel;The number of floors is six, and the storey heights are 1st storey = 4.45 m, 2nd to 4th storey = 4.10 m, 5th storey = 4.40 m, and 6th storey = 4.35 m;The storey weights are 1st storey = 15,068 kN, 2nd storey = 13,422 kN, 3rd storey = 15,290 kN, 4th storey = 9899 kN, 5th storey = 10,387 kN, and 6th storey = 11,853 kN;I cross-section and box cross-section for beams and columns, respectively;The X-direction presents an irregular configuration in its elevation (see Figure 3). Only the X-direction is analyzed in this study;The natural period (T_1_) of the building in the X-direction is 0.681 s (1.468 Hz) with an effective modal mass ratio of 0.77. The second mode period (T_2_) is 0.264 s (3.788 Hz) with an effective modal mass ratio of 0.145. The values are obtained from numerical simulations of the structural model according to Section 4.

**Figure 2 sensors-22-06426-f002:**
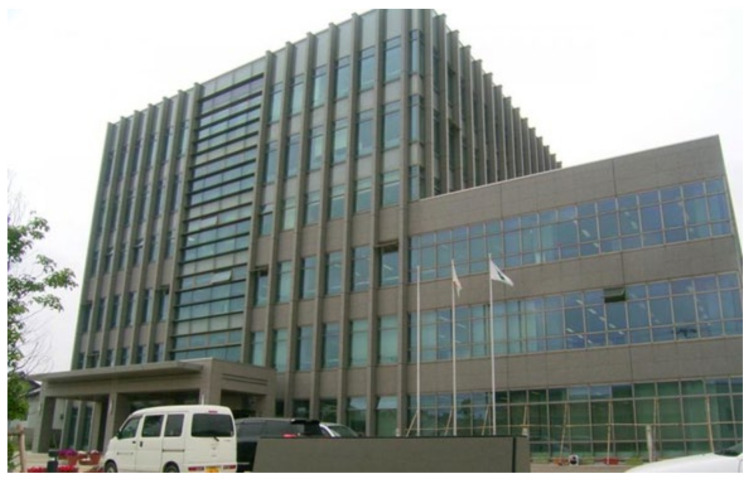
Tahara City Hall Building.

**Figure 3 sensors-22-06426-f003:**
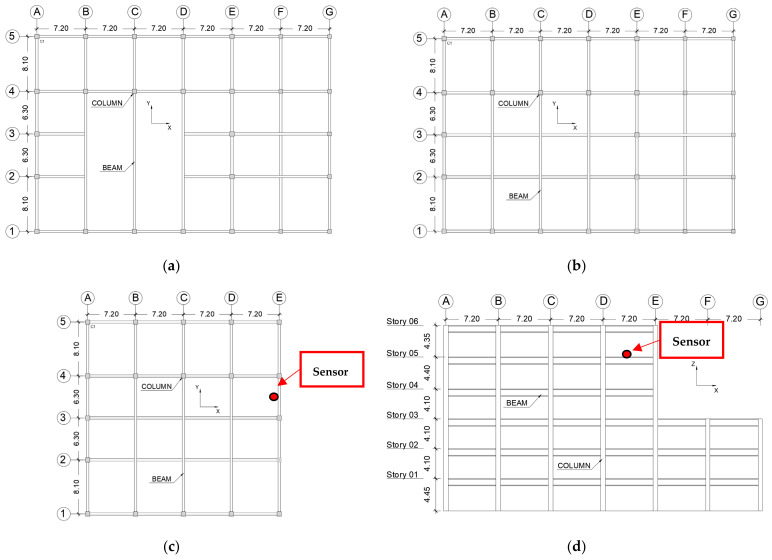
General drawings of Tahara City Hall building. (**a**) Plan of 1st storey view. (**b**) Plan of 2nd and 3rd stories’ views. (**c**) Plan from 4th to 6th storey view. (**d**) Elevation of X-direction view.

### 3.2. Toyohashi Fire Station Building

The Toyohashi Fire Station is a fire station located in Toyohashi city of Aichi prefecture in Japan (see Figure 4). This building is an instrumented building, and the location of the sensor is shown in Figure 5. The main structural characteristics are as follows:The structural system of the building is a moment-resisting frame in steel-reinforced concrete (SRC);The number of floors is six with a basement, and the typical storey height is 4.00 m;The storey weights are basement = 18,019 kN, 1st storey = 14,570 kN, 2nd storey = 12,483 kN, 3rd storey = 12,470 kN, 4th storey = 13,043 kN, 5th storey = 12,412 kN, 6th storey = 11,834 kN, and 7th storey = 10,588 kN;The steel I cross-sections are embedded in RC rectangular beams and columns;Both the X- and Y-directions are regular configurations, as shown in Figure 5. Only the X-direction is analyzed in this study;The natural period (T_1_) of the building in the X-direction is 0.748 s (1.337 Hz), with an effective modal mass ratio of 0.62. The second mode period (T_2_) is 0.277 s (3.610 Hz) with an effective modal mass ratio of 0.12. The values are obtained from numerical simulations of the structural model according to Section 4.

**Figure 4 sensors-22-06426-f004:**
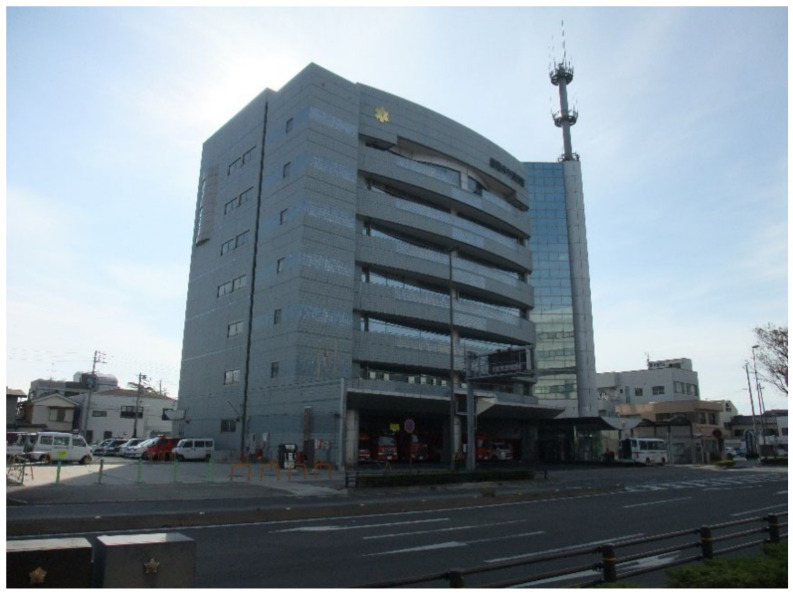
Toyohashi Fire Station Building.

**Figure 5 sensors-22-06426-f005:**
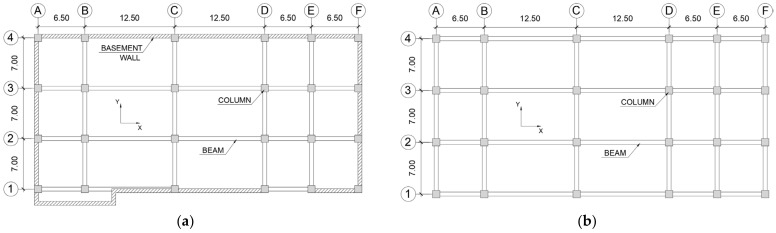
General drawings of Toyohashi Fire Station building. (**a**) Plan of basement view. (**b**) Plan from 1st to 6th storey view. (**c**) Plan of 7th storey view. (**d**) Elevation of X-direction view.

In this study, the signals of the sensors have been simulated from the 3D structural models, which were constructed element by element based on the structural drawings. After the earthquake, the sensors will be activated and read the acceleration when a threshold is reached. Subsequently, all data are automatically stored on the network cloud and can be used to assess the damage. However, for future research, it is recommended to use other methods in order to reconstruct the missing data due to anomalies and other factors [21,22].

## 4. Nonlinear Structural Models for the Target Buildings

The structural models for the target buildings consist of three-dimensional elements with elastic and inelastic behavior. The software STERA_3D [23] developed by one of the authors was used, wherein the frame beam elements were modeled using nonlinear flexural springs at their ends and a nonlinear shear spring in the middle, as shown in Figure 6.

Figure 7 shows the hysteresis models of flexural springs. Figure 7a shows the degrading trilinear slip model for the RC sections. Figure 7b shows the bilinear model for steel sections.

Likewise, the frame column elements are modeled as multi-spring models considering a nonlinear interaction between the bidirectional–flexural and axial effects (Mx-My-Nz, as shown in Figure 8a). The springs are distributed in the RC and steel cross-sections, as shown in Figure 8b. Moreover, Figure 8c shows the hysteresis model for steel and concrete springs. The nonlinear shear springs in the X- and Y-directions are defined independently.

## 5. Damage Identification of the Target Buildings

The inelastic 3D-FMs for the target buildings are made to obtain structural responses in detail. The damage condition will be identified from the maximum structural responses SD and AA on each storey.

The usability of the building is defined in order to evaluate the habitability of the building, and is represented by safe use, restricted use, and unsafe use, as shown in Table 1. The levels of damage condition shown in Table 1 are the same as in Moscoso et al. [20].

## 6. Selection of Ground Motion Records

A record database was generated from the records obtained in the Center for Engineering Strong Motion Data by USGS and the California Geological Survey. This center receives worldwide records from the cooperation of international strong-motion seismic networks [24].

In order to consider the ground motion records with high intensity and reduce the number of samples (less than 3000), the records with a PGA greater than 400 gal and a time range from 5% to 95% of the Arias intensity have been selected [25]. Finally, 183 ground motion records have been selected in this study for the database.

In the structural analysis, the variability of the structural responses of the building depends mainly on the ground motion used. On the other hand, the prediction accuracy of ML is improved when the output variability is reduced. Therefore, a methodology to select the records from structural responses has been developed using the IDA of each target building. Figure 9 shows the procedure for determining the ground motion records from the database.

For the IDA, the demand measure is either the SD or AA (on the vertical axis), and the intensity measure is the 5% damped spectral acceleration matched at the fundamental period (Sa(T_1_, 5%) on the horizontal axis). Sa(T_1_, 5%) is selected to represent the seismic intensity, where the main modal mass contribution is obtained. Besides this, a normal distribution is considered to represent the variability of the structural responses along Sa(T_1_, 5%). Thus, 68% of the structural responses are represented within ±1σ (one standard deviation) of the mean, resulting in a confidence interval from 16% to 84% fractile.

Therefore, in order to cover elastic and inelastic behavior ranges, the IDA curves use a confidence interval from 0% to 84% fractile of the structural responses. Besides this, 1/50 has been established as the maximum inter-storey drift limit. The selected records and Sa(T_1_, 5%) have been derived from accomplishing both previous conditions for SD. The IDA scale factors are such that the resultant spectral acceleration (ΔSa) is 25 gal if Sa(T_1_, 5%) max is less than 1000 gal, and 50 gal if Sa(T_1_, 5%) max is greater than 1000 gal. The IDA curves for AA, the same confidence interval, scale factors, and maximum Sa(T_1_, 5%) have been used to find their selected ground motion records. These criteria have been developed after several structural analyses carried out in this research.

### 6.1. Selection of Records

Figure 10 and Figure 11 show the IDA curves for SD and AA of the database and selected records. The black dashed lines are the 0%, 50%, and 84% fractiles. Besides this, the damage condition threshold limits are depicted by the green, yellow, orange, and red dashed lines.

Figure 12 and Figure 13 show the acceleration response spectra (Sa) of the selected records subdivided into training and validation records used for the CNN models for SD and AA analyses. Besides this, the validation records for the SD and AA analyses are the same, in order to consider the same earthquake events. Additionally, Sa is 100 gal at the fundamental period of each target building.

#### 6.1.1. Selection of Records for Tahara City Hall Building

After selecting the records, 130 and 142 records have been chosen for SD and AA, respectively, shown in Figure 10b,d.

The selected records are split into 120 and 132 for the training of CNN models for SD and AA. Besides this, ten records have been used for the validation process in both analyses.

#### 6.1.2. Selection of Records for Toyohashi Fire Station Building

After selecting the records, 115 and 144 records have been chosen for CNN models for SD and AA, respectively, shown in Figure 11b,d.

The selected records have been split into 99 and 126 for the training of CNN models for SD and AA. Besides this, 16 records are used for the validation process for both analyses.

## 7. Wavelet Power Spectrum as Input Data of CNN

The acceleration record of the upper floor is obtained from the sensor installed on the target building. Since these records are non-stationary signals, they are transformed in order to capture their characteristics in the time and frequency domains. In this study, the wavelet transform is used.

Wavelet functions convolute the original signal into a space and scale field. The scale decomposition (related to the frequency domain) is obtained by dilating and shortening the wavelet. On the other hand, space decomposition comes from their variability in time (position) [26,27].

The wavelet signal is called the mother wavelet. The wavelet used in this study is the Morlet wavelet (complex-valued wavelet), which is the product of a sine (complex exponential) wave and a Gaussian envelope, as defined by Equation (1) [27]:(1)$$\psi_{0}\left(x\right)=\pi^{-\frac{1}{4}}\cdot e^{-\frac{x^{2}}{2}}\cdot e^{-i\omega_{0}x}$$
where ω_0_ is the nondimensional frequency. In this study, ω_0_ is taken to be 6 in order to accomplish the admissibility property, according to [26]. Subsequently, $\psi_{0}$ will be normalized to keep constant the total energy when it is scaled. Furthermore, the parameters “*a*” and “*b*” are included in the Morlet wavelet in order to modify the scale and space (translation), respectively. The normalized Morlet wavelet is defined by Equation (2).
(2)$$\psi \left[\frac{\left(t-b\right)\delta t}{a}\right]={\left(\frac{\delta t}{a}\right)}^{1/2}\cdot \psi_{0}\left[\frac{\left(t-b\right)\delta t}{a}\right]$$

The continuous wavelet transform (CWT) of a discrete signal *s*_(*t*)_ is defined by Equation (3):(3)$$W\left(a,\;b\right)=\overset{N-1}{\underset{t=0}{\sum }}s_{\left(t\right)}\psi^{*}\left[\frac{\left(t-b\right)\delta t}{a}\right]$$
where “*N*” is the number of samples of the signal and the asterisk symbol (*) indicates the complex conjugate of the wavelet.

CWT is a complex function because of the Morlet wavelet. Therefore, the module of CWT is the wavelet spectrum (*WS*), defined by Equation (4), and the square of the module is the wavelet power spectrum (*WPS*), defined by Equation (5).
(4)$$WS\left(t,f\right)=\left|W\left(a,b\right)\right|=\sqrt{W_{real}^{2}+W_{imaginary}^{2}}$$
(5)$$WPS\left(t,f\right)=WS^{2}$$

*WS* was used by Moscoso et al. for the maximum absolute acceleration response on the upper floor. These results were used as images for the input data of the CNN model [20]. However, since *WPS* increases the coefficients of *WS* exponentially, the main characteristics of the signal are intensified in order to train the CNN model. For example, Figure 14 shows a random acceleration response, *WS* and *WPS* in 2D and 3D. Notice that WPS depicts the main frequencies more evidently than WS.

## 8. Convolutional Neural Network (CNN) Model

Figure 15 shows a general CNN model. The input, convolutional, pooling, and fully-connected layers compose the architecture of the CNN model in this study. Each target building has a particular CNN model architecture, which depends on its accuracy after evaluation.

The set of images is the input layer. In this study, CNN uses the images obtained from the WPS. Moreover, the convolutional layer is formed from the convolutional process, padding, and activation function methods, as shown in Figure 16.

As known, a convolutional process is a mathematical operation of two functions. The functions used in CNN are arrays of data. The set of WPS is the first array, and the second is a set of filters used to extract and learn the main features of the first one. They are called kernels (*K*) or feature detectors. Since CNN is part of the ANN, a kernel is a set of updatable weights for the training process of the CNN model.

According to the prediction project, the CNN uses one-, two-, or three-dimensional space. This study used a two-dimensional space. Equation (6) defines the convolutional operation in CNN [28]:(6)$$FM_{\left(a,b\right)}={\left(WPS\cdot K\right)}_{\left(a,b\right)}=\underset{c}{\sum }\underset{d}{\sum }K_{\left(c,d\right)}\cdot WPS_{\left(a-c,b-d\right)}$$
where *FM* is the feature map, *WPS* is the wavelet power spectrum used as input data, and *K* is the kernel array.

In order to keep the size of the original image of the feature map, the same-padding or zero-padding method is used in this study. This method adds zeros along the border of the original image. Furthermore, every value of the feature map is evaluated by the activation function method, which allows for learning the nonlinearity features of the CNN model. The rectified linear unit function (ReLU) is used in this study and is defined by Equation (7):(7)$$y=\left\{\begin{array}{c} 0,\;\;\;if\;x<0\\ x,\;\;\;if\;x\geq 0 \end{array}\right.$$

The pooling layer reduces the resolution and sizes of the feature maps, resulting in a lower computational cost. The maximum pooling layer, or max-pooling, is used for this research. The max-pooling divides the feature maps into sub-arrays with the size P × P, and the result is the maximum value of this. The CNN model without the max-pooling of Moscoso et al. converged more effectively. Nonetheless, the CNN model used for this study converged more efficiently using max-pooling layers because of the more significant amount of data. The number of the convolutional or max-pooling layer depends on the architecture of the CNN model (see Table 2).

After the convolutional and max-pooling layers, the outputs are connected to a one-dimensional array, the fully connected layer. This layer is connected to the dense layer, which provides the predicting results. The predicting results are the structural responses of each storey of the target building. As mentioned, the structural responses are used as damage indicators. It is an iterative process performed until finding the lowest Mean Squared Error (*MSE*), which is the error used in this study and defined by Equation (8). The iteration is called an “epoch”, and 50 epochs are considered in this study.
(8)$$MSE=\frac{1}{N}\cdot \sum_{i=1}^{N}{\left(y_{pred}-y_{ref}\right)}^{2},$$
where $y_{pred}$ and $y_{ref}$ are the prediction and reference results, respectively, and *N* is the number of samples. The criteria of the error measure could be modified in order to improve the forecasting accuracy [29].

The hyperparameters used in this study are shown in Table 2. They were obtained after several processes of training and validation. However, methods are recommended to optimize the hyperparameters [30,31].

## 9. Training and Validation Processes

As mentioned in the methodology (Section 2), there are two processes used to obtain a trained CNN model, the training process (TP) and the validation process (VP).

In the TP, the CNN model is trained using the WPS of the absolute acceleration on the top floor of the target building. Then, the prediction results of the CNN model are compared to the SD and AA of each floor of the target building. The IDA numerical procedure is called the data preparation process in the methodology, and obtains the SD and AA. The TP provides a prepared model to make predictions; however, its accuracy should be checked in the validation process.

In the VP, new input data are obtained by the data preparation process. The new WPS is used in the trained CNN model and automatically predicts the results (SD or AA). The following results are compared to reference data:The usability of the building, in which the availability of the building occupancy is evaluated after an earthquake,The total damage condition, in which it is possible to identify the damage state of the target building,Storey damage condition, in which it is possible to identify the damage state of each floor of the target building,Total comparison of the SD or AA.

In general, one of the most potent advantages of the ML method in SHM is the rapid prediction result when an earthquake occurs. In other words, even though the TP and VP take a long time to obtain the final CNN model, it is carried out before the earthquake, and the prediction is obtained automatically. Therefore, it is possible to identify the damage states of actual buildings (3D regular or irregular structural configurations) immediately after the earthquake.

## 10. Prediction Results of Target Buildings

The results of predicting the responses and damage levels of the target buildings are summarized as follows:Figure 17 and Figure 18 show SD and AA results of the Tahara City Hall building, respectively;Figure 19 and Figure 20 show SD and AA results of the Toyohashi Fire Station building, respectively;Figure 17a, Figure 18a, Figure 19a and Figure 20a show the training loss using MSE, which decreases with the epochs increasing in TP;A confusion matrix is used to evaluate the prediction accuracy of the total and storey damage condition (see Figure 17b,c, Figure 18b,c, Figure 19b,c and Figure 20b,c). The confusion matrix represents the correct and incorrect predictions through the number of coincidences with the reference data. The rows and columns of the matrix are tagged as the predicted and the true label, respectively. Therefore, the number of well-matched predictions is located on the diagonal of the matrix.Figure 17d, Figure 18d, Figure 19d and Figure 20d show the accuracy of the damage condition of each floor.Figure 17e, Figure 18e, Figure 19e and Figure 20e show the comparison of the prediction results using the coefficient of determination or *R*-squared as defined by Equation (9):
(9)$$R^{2}=1-\frac{\sum_{i}^{N}{\left(y_{ref,\;i}-y_{pred,\;i}\right)}^{2}}{\sum_{i}^{N}{\left(y_{ref,\;i}-\bar{y_{ref}}\right)}^{2}}$$
where $y_{pred}$ and $y_{ref}$ are the prediction and reference results, respectively. Furthermore, $\bar{y_{ref}}$ is the mean of the reference values and *N* is the number of samples.

Table 3 shows the evaluation accuracy of the target buildings in the VP. The results are summarized below:For the Tahara City Hall building, the maximum accuracy and R^2^ are 90.0% (usability of the building) and 0.825, respectively;For the Toyohashi Fire Station building, the maximum accuracy and R^2^ are 100% (damage condition of the basement) and 0.909, respectively;In general, the accuracy of the estimation of SD is the highest.

**Table 3 sensors-22-06426-t003:** Evaluation accuracy of target buildings.

Accuracy Evaluation		Tahara City Hall Building		Toyohashi Fire Station Building	
		SD	AA	SD	AA
Usability of the building (Accuracy)		90.0%	84.2%	94.1%	88.1%
Total damage condition (Accuracy)		76.1%	74.5%	82.2%	71.2%
Storey damage condition (Storey accuracy)	Basement	--	--	100%	60.0%
	Storey 1	76.8%	58.7%	96.9%	63.6%
	Storey 2	76.9%	64.4%	94.1%	65.3%
	Storey 3	76.3%	69.2%	92.4%	65.8%
	Storey 4	74.8%	71.3%	91.1%	65.2%
	Storey 5	74.4%	71.4%	89.9%	63.0%
	Storey 6	75.3%	71.9%	88.8%	62.7%
	Storey 7	--	--	87.8%	62.7%
Total comparison (R^2^)		0.825	0.817	0.909	0.732

## 11. Conclusions and Discussion

In this research article, a previous methodology proposed by the authors has been improved and applied to two instrumented buildings in Aichi Prefecture in Japan, called Tahara City hall and Toyohashi Fire Station. The summary of the proposed methodology is as follows:CNN models are trained per target building using the WPS of the absolute acceleration of the top floor record as input data to predict the SD and AA values. SD and AA are used as indicators to detect the damage state of the structures;A methodology to select records in order to reduce the variability of the structural responses using IDA is proposed, wherein the confidence interval between the 0% and 84% fractiles is adopted;The evaluation accuracy is discussed on the usability of the building, total damage condition, storey damage condition, and total comparison of the damage indicator;The maximum accuracy and R^2^ for the Tahara City Hall building are 90.0% (usability of the building) and 0.825, respectively;The maximum accuracy and R^2^ for the Toyohashi Fire Station building are 100% (damage condition of the basement) and 0.909, respectively;In general, the accuracy of the estimation of SD is the highest.

Finally, the improved methodology based on CNN immediately detects the structural damage condition of buildings, considering only one sensor on the top floor. Since the training and validation process are computed before, a prediction can be obtained immediately after an earthquake.

## Figures and Tables

**Figure 1 sensors-22-06426-f001:**
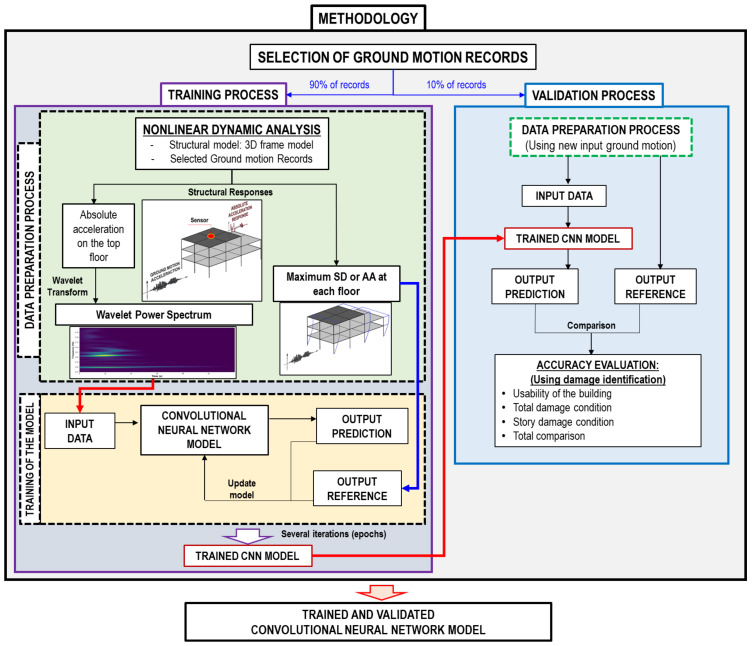
Methodology flowchart to obtain the Convolutional Neural Network model.

**Figure 6 sensors-22-06426-f006:**
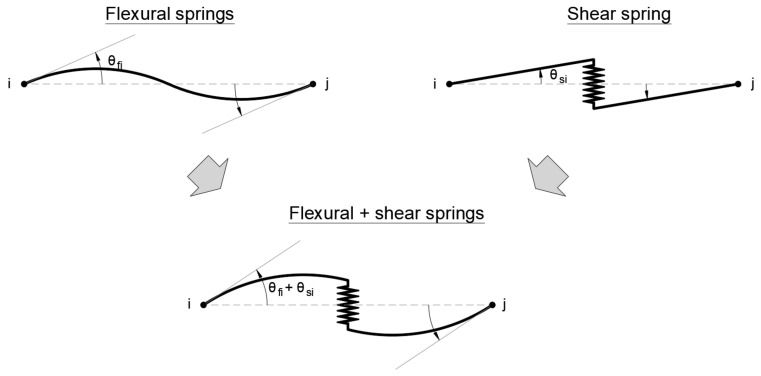
Beam model with nonlinear flexural and shear springs [23].

**Figure 7 sensors-22-06426-f007:**
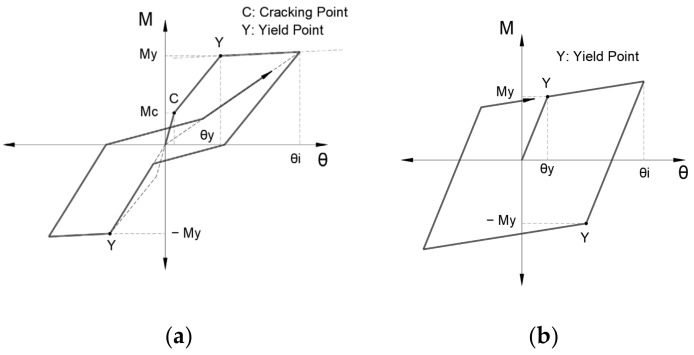
Hysteresis model. (**a**) Degrading trilinear slip model for RC sections; (**b**) bilinear model for steel sections [23].

**Figure 8 sensors-22-06426-f008:**
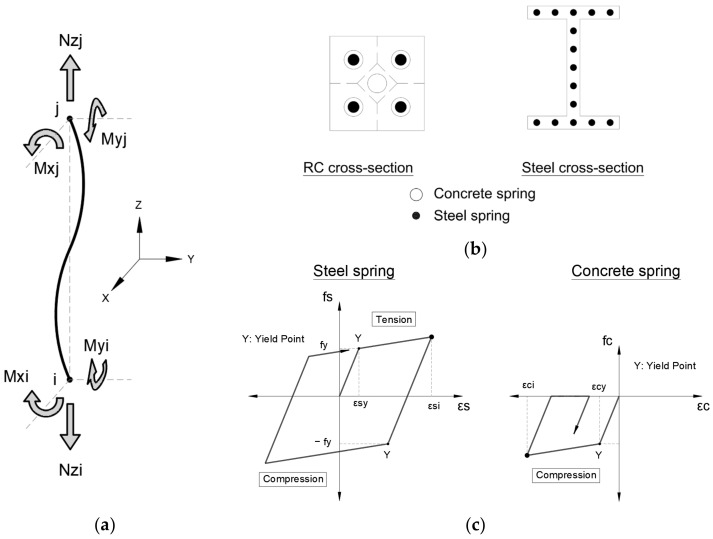
(**a**) Column model with multi-springs to consider Nz-Mx-My nonlinear interaction; (**b**) concrete and steel springs; (**c**) hysteresis model for steel and concrete springs [23].

**Figure 9 sensors-22-06426-f009:**
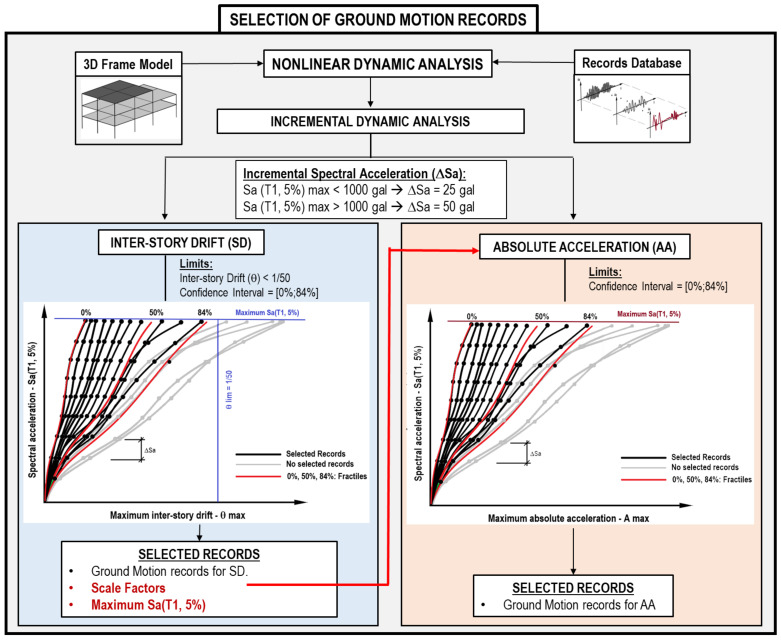
Selection of ground motion records for inter-storey drift and acceleration flowchart.

**Figure 10 sensors-22-06426-f010:**
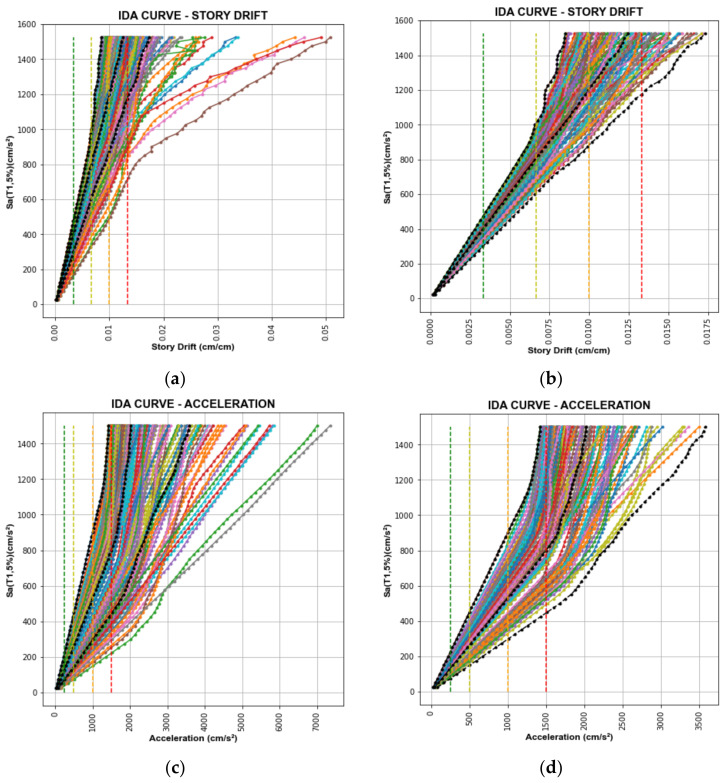
IDA Curves of Tahara City Hall Building. (**a**) IDA curves of the database for SD; (**b**) IDA curves of selected records for SD; (**c**) IDA curves of the database for AA; (**d**) IDA curves of selected records for AA.

**Figure 11 sensors-22-06426-f011:**
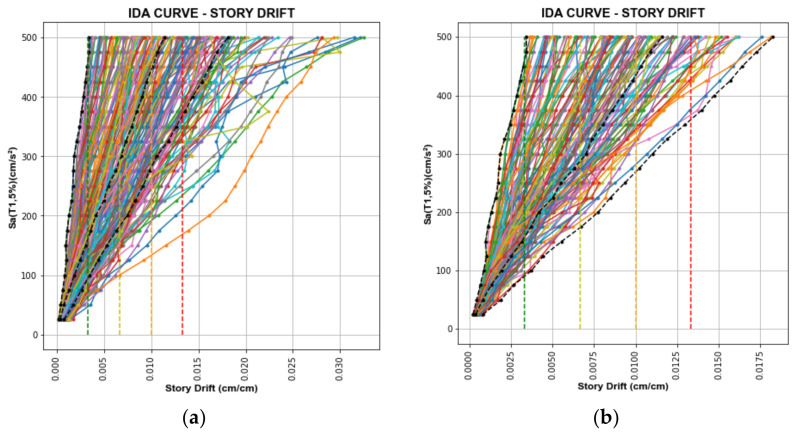
IDA Curves of Toyohashi Fire Station Building. (**a**) IDA curves of the database for SD; (**b**) IDA curves of selected records for SD; (**c**) IDA curves of the database for AA; (**d**) IDA curves of selected records for AA.

**Figure 12 sensors-22-06426-f012:**
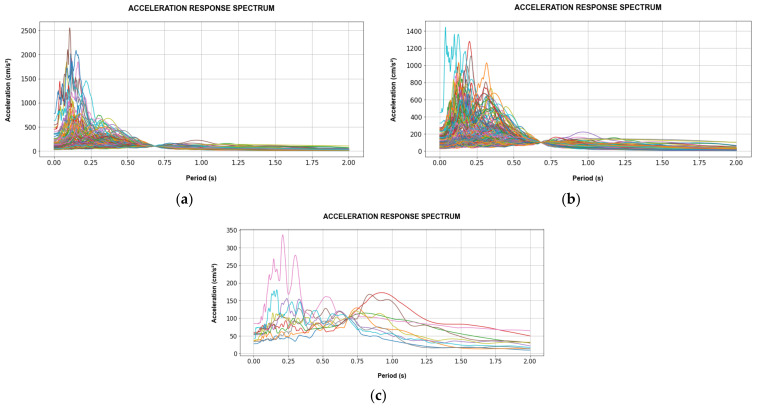
Acceleration response spectrum of Tahara City Hall Building at T_1_ = 0.681 s and Sa(T_1_) = 100 gal. (**a**) Training records for SD; (**b**) Training records for AA; (**c**) Validation records for SD and AA analyses.

**Figure 13 sensors-22-06426-f013:**
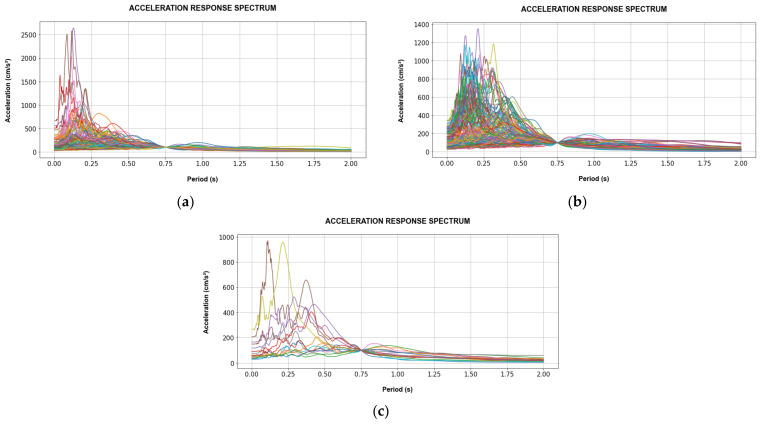
Acceleration response spectrum of Toyohashi Fire Station Building at T_1_ = 0.748 s and Sa(T_1_) = 100 gal. (**a**) Training records for SD; (**b**) training records for AA; (**c**) validation records for SD and AA analyses.

**Figure 14 sensors-22-06426-f014:**
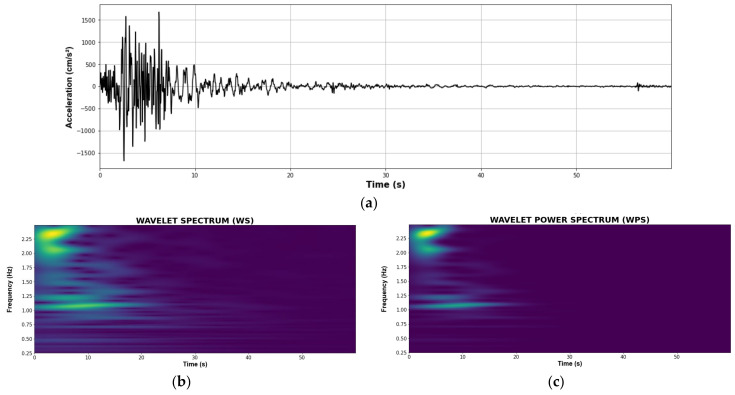
(**a**) Original signal; (**b**) two-dimensional WS; (**c**) two-dimensional WPS; (**d**) three-dimensional WPS; (**e**) three-dimensional WPS.

**Figure 15 sensors-22-06426-f015:**
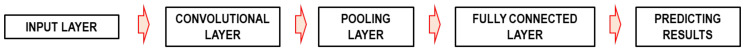
A general CNN model.

**Figure 16 sensors-22-06426-f016:**
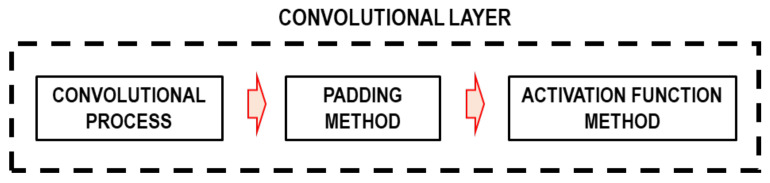
Convolutional layer.

**Figure 17 sensors-22-06426-f017:**
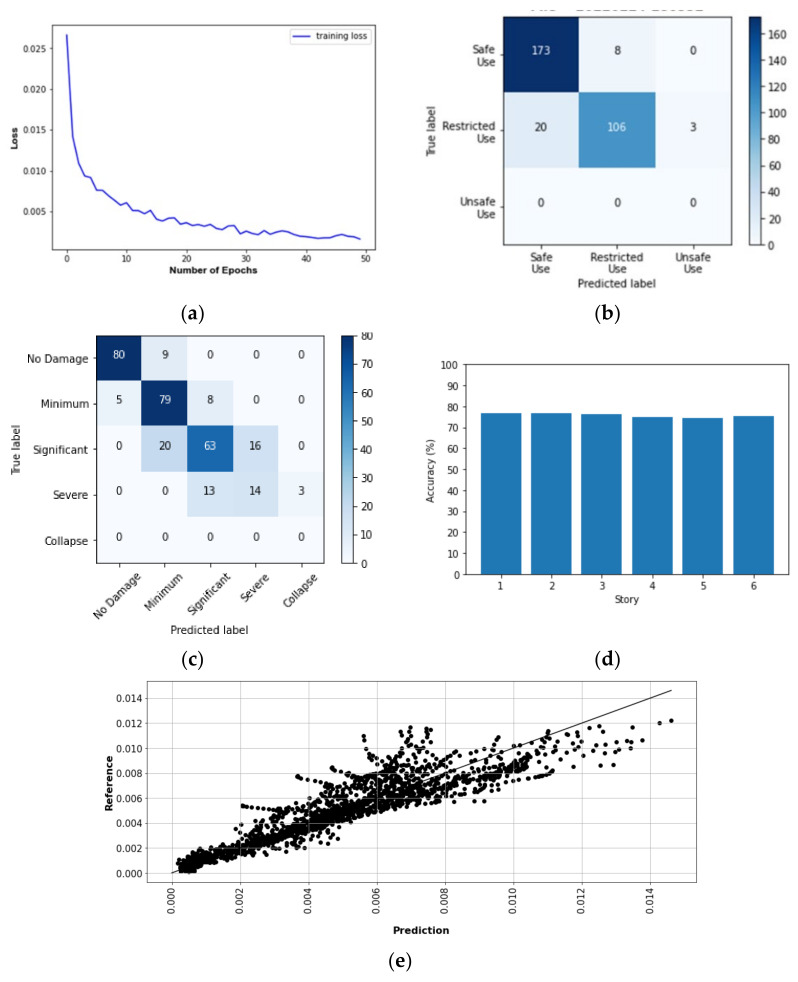
SD results of the TP and VP for Tahara City Hall building: (**a**) Convergence curve—loss in the TP; (**b**) confusion matrix—usability of the building by VP; (**c**) confusion matrix—total damage condition by VP; (**d**) confusion matrix—storey damage condition by VP; (**e**) total comparison of SD.

**Figure 18 sensors-22-06426-f018:**
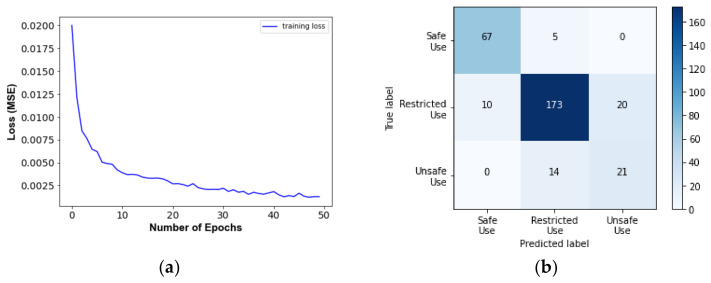
AA results of the TP and VP for Tahara City Hall building: (**a**) convergence curve—loss in the TP; (**b**) confusion matrix—usability of the building by VP; (**c**) confusion matrix—total damage condition by VP; (**d**) confusion matrix—storey damage condition by VP; (**e**) total comparison of AA.

**Figure 19 sensors-22-06426-f019:**
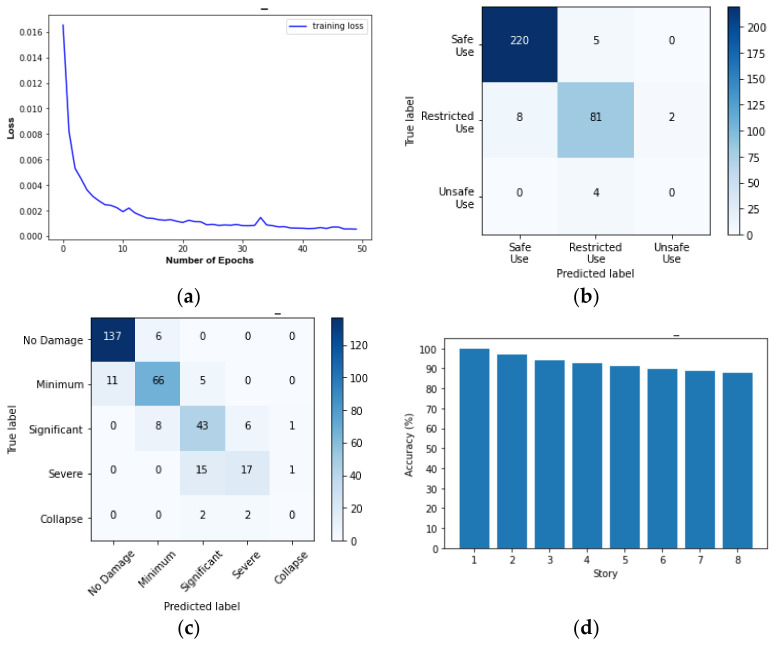
SD results of the TP and VP for Toyohashi Fire Station building: (**a**) convergence curve—loss in the TP; (**b**) confusion matrix—usability of the building by VP; (**c**) confusion matrix—total damage condition by VP; (**d**) confusion matrix—storey damage condition by VP; (**e**) total comparison of SD.

**Figure 20 sensors-22-06426-f020:**
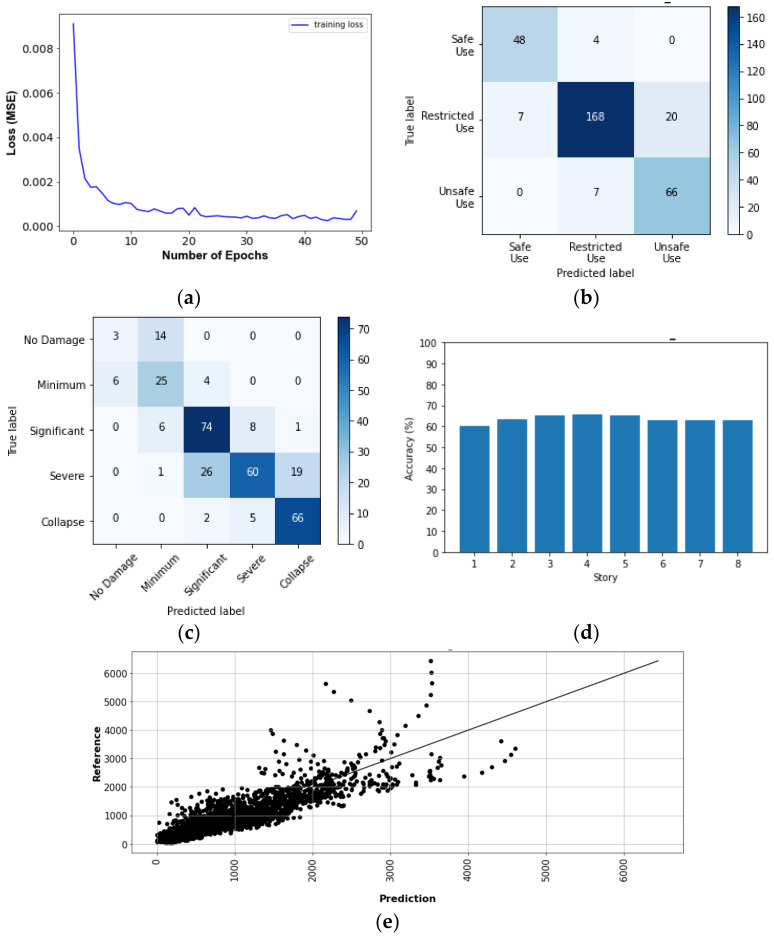
AA results of the TP and VP for Toyohashi Fire Station building: (**a**) convergence curve—loss in the TP; (**b**) confusion matrix—usability of the building by VP; (**c**) confusion matrix—total damage condition by VP; (**d**) confusion matrix—storey damage condition by VP; (**e**) total comparison of AA.

**Table 1 sensors-22-06426-t001:** Damage conditions from the structural responses [20].

Usability of the Building	Safe Use		Restricted Use	Unsafe Use	
Damage Condition	No Damage	Minimum Damage	Significant Damage	Severe Damage	Collapse
Inter-storey drift ratio	<1/300	≥1/300 but <1/150	≥1/150 but <1/100	≥1/100 but < 1/75	≥1/75
Acceleration (gal)	<250	≥250 but <500	≥500 but <1000	≥1000 but < 1500	≥1500

**Table 2 sensors-22-06426-t002:** Hyperparameters of CNN models.

Layer	Type	Hyperparameter	Tahara City Hall Building		Toyohashi Fire Station Building	
			SD	AA	SD	AA
01	Convolutional	Number of kernels	8	8	8	8
		Size of kernels	3 × 3	3 × 3	3 × 3	3 × 3
02	Pooling	Size of pooling filter	2 × 2	2 × 2	2 × 2	2 × 2
03	Convolution	Number of kernels	8	8	8	8
		Size of kernels	3 × 3	3 × 3	3 × 3	3 × 3
04	Pooling	Size of pooling filter	2 × 2	2 × 2	---	2 × 2
05	Convolution	Number of kernels	8	8	8	8
		Size of kernels	3 × 3	3 × 3	3 × 3	3 × 3
06	Pooling	Size of pooling filter	2 × 2	2 × 2	---	2 × 2
07	Convolution	Number of kernels	8	8	8	8
		Size of kernels	3 × 3	3 × 3	3 × 3	3 × 3
08	Pooling	Size of pooling filter	2 × 2	2 × 2	2 × 2	2 × 2
09	Fully connected	Output	6	6	7	7

## Data Availability

The data presented in this study are available on request from the corresponding author.

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
