# Peer review of "Convolutional Neural Network-Based Rapid Post-Earthquake Structural Damage Detection: Case Study"

_sensors, 2022, doi:10.3390/s22176426_

Round 1

Reviewer 1 Report

This paper presents the rapid post-earthquake structural damage detection for two buildings using the CNN. The wavelet power spectrum is used as the input variables, and the maximum inter-story drift and absolute acceleration of each story are used as the damage indicators. The subject is meaningful and interesting, but the following comments should be carefully considered to improve this paper.

1. The hyperparameters are not given for CNN. Which method do you use to determine them? It is recommended to use Bayesian optimization to determine the hyperparameters. More details can refer to Probabilistic framework with Bayesian optimization for predicting typhoon-induced dynamic responses of a long-span bridge.

2. The pre-processing of data is important, as the measurements inevitably involve anomalies, such as outliers and missing data, which can directly affect the reliability of the subsequent analysis. However, the paper did not give the details. In particular, the Bayesian methods with high accuracy are recommended to be briefly reviewed, and the associated reference like Bayesian dynamic regression for reconstructing missing data in structural health monitoring could be added.

3. It is seen from Figs. 18 and 19 that the loss decreases until the epoch reaches 50. Thus, it is suggested to increase the epoch to get more reliable results.

4. The authors should use different criteria to quantitively compare the forecasting results by a different model. The mean absolute percent error (MAPE), mean absolute error (MAE) and mean bias error (or Bias) are recommended. The three criteria can be referred to Sparse Gaussian process regression for multi-step ahead forecasting of wind gusts combining numerical weather predictions and on-site measurements. This reference should be added to present the above criteria.

Reviewer 2 Report

This manuscript focused on a CNN methodology to detect the structural damage condition of a building. In general, the manuscript gives a good description of the proposed methodology, and the data analysis and results are fully discussed. However, the manuscript needs to be revised, and the analysis presented in the manuscript is somehow limited and required to be extended. The detailed comments: 

1. It is necessary to explain clearly the superiority of the proposed method through comparison with other methods.

2. It is mentioned in the paper that CNN training is slow, but the obtained model prediction is fast. What is the specific training speed and inference speed? What are the advantages compared to non-CNN models?

3. There are too many chapters in this manuscript, which leads to that the key points are not well highlighted. Please improve it significantly.

Reviewer 3 Report

This manuscript presents a Convolutional Neural Network-based method to detect the structural damage condition of buildings after an earthquake. The presented method is an improvement of a damage identification methodology previously proposed by the authors. Approaching two case studies, the authors demonstrate that the methodology can detect sufficiently precisely the damaged state of the structures.

Overall, the manuscript is well written and interesting. However, some minor recommendations are listed below.

1. Is there any evidence about how accurate the data obtained from numerical simulations of the structural model are?

2. How was the location of the sensor in the two buildings chosen?

3. Would it be possible to include more information about how the acceleration record of the upper floor is obtained from the sensor?

4. How could the proposed methodology be applied in a real case?

Round 2

Reviewer 1 Report

The authors have addressed my previous comments.